# BizFinBench: A Business-Driven Real-World Financial Benchmark for Evaluating LLMs

## Abstract

Large language models excel in general tasks, yet assessing their reliability in logic-heavy, precision-critical domains like finance, law, and healthcare remains challenging. To address this, we introduce BizFinBench, the first benchmark specifically designed to evaluate LLMs in real-world financial applications. BizFinBench consists of 7,605 well-annotated queries in Chinese, spanning five dimensions: numerical calculation, reasoning, information extraction, prediction recognition, and knowledge-based question answering, grouped into nine fine-grained categories. The benchmark includes both objective and subjective metrics. We also introduce IteraJudge, a novel LLM evaluation method that reduces bias when LLMs serve as evaluators in objective metrics. We evaluate 30 models, covering both proprietary and open-source systems. The results highlight several key trends: (1) **Numerical Calculation**: GPT-5 and Gemini-2.5-Pro achieve the best performance, while the open-source DeepSeek-v3.1 demonstrates substantial progress, narrowing the gap with proprietary leaders; (2) **Reasoning**: proprietary models retain a clear advantage, outperforming open-source counterparts by approximately 10.74%; (3) **Information Extraction**: DeepSeek-R1 and DeepSeek-V3 deliver competitive results, closely approaching GPT-5 and Gemini-2.5-Pro; (4) **Prediction Recognition**: reasoning models (e.g., OpenAI o3 and o4-mini) achieve superior performance. Overall, no single model exhibits dominance across all dimensions, underscoring the multifaceted challenges of financial reasoning. We find that while current LLMs handle routine finance queries competently, they struggle with complex scenarios requiring cross-concept reasoning. BizFinBench offers a rigorous, business-aligned benchmark for future research. The code and dataset are included in the supplementary material and will be released publicly upon acceptance.

## 1 Introduction

Recent years have witnessed rapid advancements of Large Language Models (LLMs), which demonstrates remarkable capabilities across diverse domains, such as finance, law, healthcare and so on Chen et al. (2024); Liu et al. (2025); Zhang et al. (2023a); Lu et al. (2024); Xie et al. (2025). In financial applications, LLMs are increasingly applied to complex tasks, including automated financial analysis, fraud detection, risk assessment, and investment strategy formulation Zhao et al. (2024); Gan et al. (2024). However, evaluating the robustness and reliability of LLMs in finance domains remains a significant challenge.

Different from traditional Science, Technology, Engineering, and Mathematics (STEM) questions, where inputs are typically short, well-structured, and yield deterministic answers, financial tasks are more complex. They typically involve long context, structured inputs (e.g., tabular stock data, market news), require temporal reasoning, and demand fine-grained judgment under ambiguity. As illustrated in Figure 1, STEM-style questions usually have clear computational logic and a single correct answer, while financial tasks call for multi-step reasoning over real-world data, generally with adversarial or noisy context Du et al. (2024).

Despite the emergence of financial benchmarks such as FinEval Zhang et al. (2023b), existing approaches treat financial tasks as general document Query-Answering (QA) Wang et al. (2024),

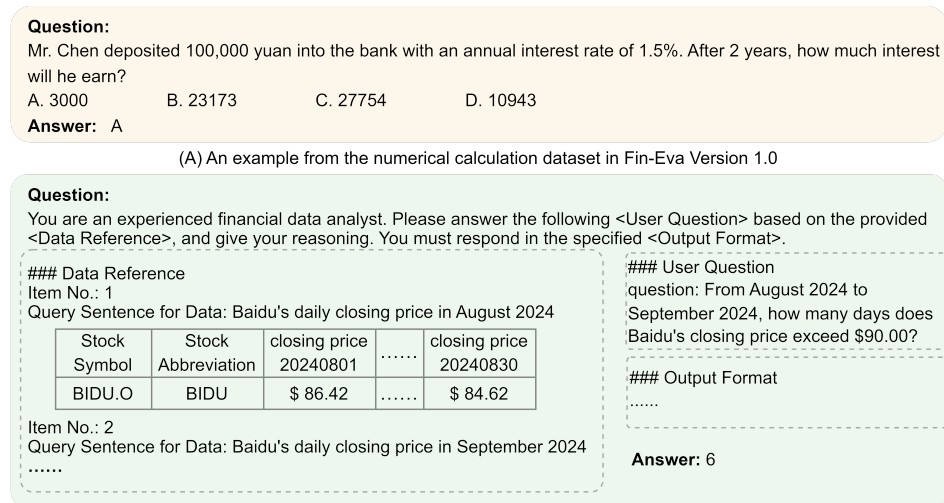

Figure 1: Comparison of numerical calculation questions in Fin-Eva Team (2023) and BizFinBench. The Fin-Eva example presents a straightforward financial math problem, while the BizFinBench example requires multi-step reasoning: first analyzing the problem, then extracting and utilizing relevant data from a provided markdown-formatted table for accurate computation. An Chinese version is included in the Appendix for clarity and ease of reference.

lacking structured inputs and business-grounded reasoning required in practice. Thus, there emerges the gap between benchmark performance and real-world applicability.

To address these limitations, we introduce BizFinBench, a comprehensive benchmark designed to rigorously evaluate LLMs across a broad spectrum of real-world financial tasks. In contrast to previous benchmarks, BizFinBench adopts a business-driven data construction methodology and emphasizes contextual complexity and adversarial robustness. It encompasses five key dimensions: QA, prediction & recognition, reasoning, information extraction, and numerical calculation. Under these dimensions, BizFinBench comprises nine distinct categories: anomalous event attribution, financial numerical computation, financial time reasoning, financial tool usage, financial knowledge QA, financial data description, emotional value evaluation, stock price prediction, and financial named entity recognition.

A core characteristic of BizFinBench is the focus on business-contextual evaluation. For example, in the anomalous event attribution task, LLMs are required to identify the causes of stock price anomalies by analyzing time-sensitive news feeds, some of which are deliberately embedded with misleading positive or negative information. This setting challenges LLMs to perform fine-grained reasoning and signal discrimination under realistic noise and uncertainty.

In addition to the benchmark design, a critical component of BizFinBench is the design of a reliable evaluation methodology. While constructing realistic tasks is essential, evaluating LLM outputs, particularly for open-ended, complex financial problems, remains a significant challenge.

Table 1: Comparison Between BizFinBench and Other Financial Datasets

| Data | Year | Task | Examples | Language | Source | Business-based |
|---|---|---|---|---|---|---|
| FLUE | 2022 | Multiple financial NLP tasks | 26292 | English | Aggregated from existing sources | ✗ |
| FLARE | 2023 | Multiple financial NLP tasks, financial prediction tasks | 19196 | Chinese, English | Aggregated from existing sources | ✗ |
| CF-Benchmark | 2024 | Multiple financial NLP tasks | 3917 | Chinese | except | ✗ |
| FinEval | 2023 | Multiple financial NLP tasks | 8351 | Chinese | Financial field examination & except | ✗ |
| FinQA | 2021 | Financial numerical reasoning | 8281 | English | except | ✗ |
| FinanceIQ | 2023 | Multiple financial NLP tasks | 7137 | Chinese | Financial field examination & except | ✗ |
| CGCE | 2023 | Multiple financial NLP tasks | 150 | Chinese, English | except | ✗ |
| CFLUE | 2024 | Multiple financial NLP tasks | 54000 | Chinese | Financial field examination & except | ✗ |
| **BizFinBench** | 2025 | Multiple financial NLP tasks, financial prediction tasks | 7605 | Chinese | except | ✓ |

Traditional human evaluation provides high-quality judgments but suffers from two major drawbacks: (1) the annotation cost increases exponentially with the scale and domain specificity of financial tasks, and (2) subjective inconsistencies among annotators can introduce substantial noise. Although recent approaches like LLM-as-a-Judge Gu et al. (2024) attempt to automate evaluation through prompt-based simulations of human judgment, they are prone to prompt bias and generally lack alignment with expert-level assessments. These limitations are further magnified in the financial domain, where tasks demand multi-step reasoning, contextual interpretation, and robustness against adversarial or misleading signals. As such, existing evaluation paradigms are insufficient to capture the depth and nuance required for trustworthy assessment.

To address this gap, we propose *IteraJudge*, an iterative calibration-based evaluation framework tailored for financial LLM benchmarks. Drawing inspiration from the RevisEval framework Zhang et al. (2024a), *IteraJudge* enhances evaluation accuracy and reliability through three core mechanisms: evaluation dimension disentanglement, sequential correction generation, and reference-aligned assessment. By integrating *IteraJudge* into BizFinBench, we establish a rigorous and interpretable evaluation pipeline for LLM performance in high-stakes financial contexts.

In summary, the major contributions of our work are as follows:

- We propose BizFinBench, the first evaluation benchmark in the financial domain that integrates business-oriented tasks, covering 5 dimensions and 9 categories. It is designed to assess the capacity of LLMs in real-world financial scenarios.

- We design a novel evaluation method, i.e., IteraJudge, which enhances the capability of LLMs as a judge by refining their decision boundaries in specific financial evaluation tasks.

- We conduct a comprehensive evaluation with 30 LLMs based on BizFinBench, uncovering key insights into their strengths and limitations in financial applications.

## 2 RELATED WORK

In this section, we present existing evaluation benchmarks in financial domains. Then, we present the major LLMs specialized in financial domains.

### 2.1 FINANCIAL EVALUATION BENCHMARKS

FLUE Shah et al. (2022) is a comprehensive suite of benchmarks covering five key financial tasks: sentiment analysis, news headline classification, named entity recognition, structural boundary detection, and question answering. Building on FLUE, FLARE Xie et al. (2023) expands the evaluation to include time-series processing capabilities, adding tasks such as stock price movement prediction.

In addition to FLUE and FLARE, several specialized datasets focus on various aspects of financial evaluation. For example, FinQA Chen et al. (2022a) provides QA pairs annotated by financial experts, accompanied by earnings reports from S&P 500 companies. This dataset supports financial question answering, emphasizing detailed, factual responses based on corporate financial data. ConvFinQA Chen et al. (2022b) extends this by incorporating multi-turn dialogues, enabling more sophisticated interactions within the context of earnings reports, thus broadening the scope of financial evaluation to conversational contexts.

FinEval Zhang et al. (2023b) adopts a quantitative evaluation approach, combining long-term research insights with manual curation and featuring diverse question types. However, it primarily emphasizes static knowledge assessment and lacks coverage of dynamic, real-time financial tasks and fine-grained capability diagnostics, which limits its effectiveness in benchmarking models under complex, business-driven financial scenarios.

Expanding beyond traditional financial instruments such as stocks, bonds, and mutual funds, FinancelQ Duxiaoman DI Team (2023) introduces emerging topics such as cryptocurrencies and blockchain technologies. This dataset can be exploited for evaluating models in the rapidly evolving field of digital finance.

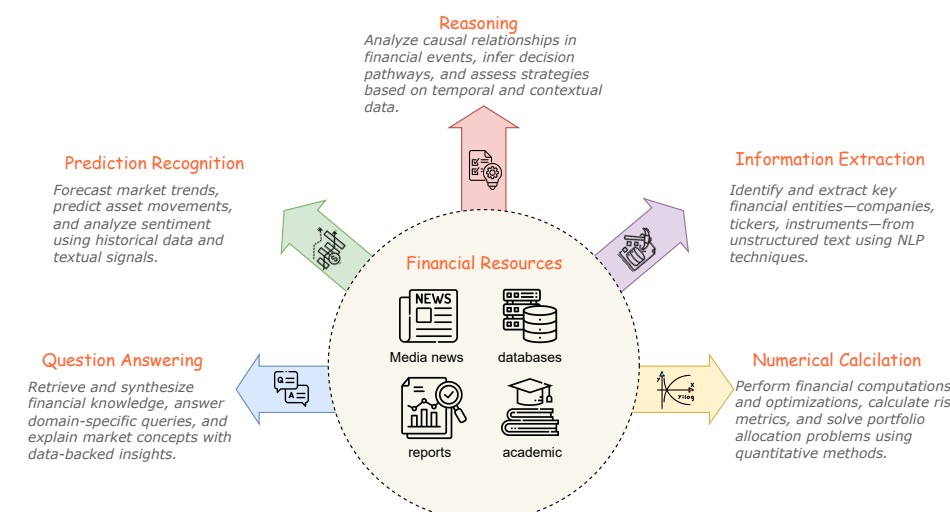

Figure 2: Distribution of tasks in BizFinBench across five key dimensions. The benchmark is structured around five dimensions, each focusing on a distinct capability of financial large language models. The figure also briefly illustrates the core focus of each dimension.

In the context of Chinese financial benchmarks, several recent datasets have been released, including CFBenchmark Lei et al. (2023), which focuses on Chinese financial text analysis; DISC-FINSFT Chen et al. (2023), designed for financial sentiment analysis and forecasting; CGCE Zhang et al. (2023c), which extends financial evaluation to include general knowledge and commonsense reasoning in Chinese financial documents; and CFLUE Jie Zhu (2024), which incorporates diverse financial examination questions and covers five fundamental NLP tasks, thereby enabling systematic and rigorous assessment of Chinese financial language understanding.

Table 1 provides a comprehensive comparison of existing financial benchmarks, detailing key aspects such as the year of release, the number of samples, language coverage, data sources, and whether the dataset was constructed with real business scenarios in mind. From the comparison, it is evident that while several benchmarks focus on financial knowledge or specific task types, they often rely on synthetic data or public information without a strong connection to actual business applications. In contrast, BizFinBench is the only benchmark explicitly designed around real-world financial operations and user interactions, making it uniquely positioned to evaluate the practical effectiveness of LLMs in authentic business environments. This business-centric design ensures higher relevance, realism, and applicability of the tasks included in the benchmark.

## 2.2 FINANCIAL LARGE LANGUAGE MODELS

By training on a large corpus of financial data based on BERT, FinBERT Araci (2019) was proposed as a pre-trained model for the financial domain, primarily used for sentiment analysis of financial texts. Subsequently, models such as FinMA Xie et al. (2023), InvestLM Yang et al. (2023a), and FinGPT Yang et al. (2023b) were fine-tuned on LLaMA Touvron et al. (2023) to further enhance their performance in the financial domain. The XuanYuan3-70B model, built on the LLaMA3-70B architecture and incrementally pre-trained with a vast amount of Chinese and English corpora, focuses on the financial sector and is capable of handling complex tasks such as financial event interpretation, investment research applications, compliance, and risk management. BloombergGPT Wu et al. (2023) is a 50-billion-parameter LLM based on the Bloom architecture, specifically designed for the financial industry, demonstrating strong adaptability in the financial domain. Meanwhile, Baichuan4-Finance Zhang et al. (2024b) has achieved an accuracy rate of over 95% in various certification fields such as banking, insurance, funds, and securities, further proving its exceptional performance in the vertical financial sector. Dianjin-R1 Zhu et al. (2025) is designed for complex financial reasoning tasks and incorporates structured supervision along with dual-reward reinforcement learning, enabling it to outperform strong baselines across a range of financial benchmarks.

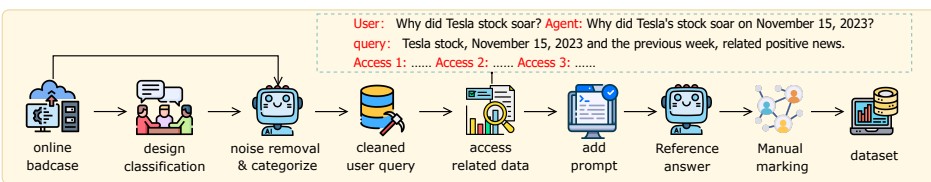

Figure 3: Workflow of BizFinBench dataset construction.

In addition, we also consider general-purpose models in our experiments, as many of them have undergone pre-training on datasets that contain financial texts, such as GPT-4o.

## 3 BIZFINBENCH

In this section, we detail the design of BizFinBench, a comprehensive benchmark specialized for evaluating LLMs in financial domains. Compared to previous datasets, BizFinBench places a strong emphasis on business practicality and real-world applicability, aiming to bridge the gap between academic evaluation and the complex challenges encountered in real-world financial scenarios.

To capture the multifaceted nature of financial intelligence, we organize the benchmark into 9 distinct task types, which are further grouped into 5 overarching evaluation dimensions. As illustrated in Figure 2 [1], these dimensions reflect key capabilities required in financial applications. For instance, the numerical computation dimension includes tasks that require models to perform financial computations and optimizations, calculate risk metrics, and solve portfolio allocation problems using quantitative methods. This dimension is designed to evaluate the capability of LLMs to apply precise mathematical reasoning in realistic financial contexts, where accuracy and analytical rigour are critical. This structured categorization not only facilitates a fine-grained assessment of model strengths and weaknesses but also ensures that each component of the benchmark aligns with practical demands observed in financial services and business analytics.

### 3.1 DATA CONSTRUCTION

Our dataset is primarily derived from real user queries on Platform A[2]. The platform serves a large community of retail investors and financial professionals, offering functionalities such as stock screening, market analysis, and personalized investment assistance. By leveraging advanced AI technologies, Platform A enables users to perform complex financial analyses via natural language queries, covering domains such as A-shares, Hong Kong and U.S. stocks, ETFs, and macroeconomic indicators.

Through a systematic analysis of user queries, financial experts identified nine representative task categories frequently encountered in real-world scenarios (e.g., temporal reasoning, numerical computation, sentiment analysis). Together, these categories account for over 90% of all queries observed on the platform, underscoring their representativeness of practical financial decision-making needs.

To construct the dataset, we first aggregated a large pool of authentic queries. We then employed GPT-4o OpenAI (2023) to filter noisy entries and classify valid queries into the expert-defined categories. For each query, we retrieved relevant contextual data from internal financial databases and external sources, including stock prices, trading history, financial news, and company disclosures. Context retrieval was anchored to the original query timestamp to ensure temporal consistency. For example, when a user asked *"Why did Tesla stock soar?"*, the query itself lacked explicit temporal markers. By aligning the query with its issue time, we retrieved financial information and news specifically associated with that period.

To increase the dataset's discriminative power, we deliberately introduced distractor information into the context. These distractors include plausible but irrelevant or misleading data, such as unrelated

---

[1]An English version is included in the Appendix J

[2]In compliance with the double-blind review policy, the actual name will be disclosed in the final version.

Table 2: Overview of BizFinBench Datasets

| Category | Data | Evaluation Dimensions | Metrics | Numbers | Avg Len. |
|---|---|---|---|---|---|
| Reasoning | Anomalous Event Attribution (AEA) | Causal consistency Information relevance Noise resistance | Accuracy | 1888 | 27902 |
| | Financial Time Reasoning (FTR) | Temporal reasoning correctness | Accuracy | 514 | 1162 |
| | Financial Tool Usage (FTU) | Tool selection appropriateness Parameter input accuracy Multi-tool coordination | Judge Score | 641 | 4556 |
| Numerical calculation | Financial Numerical Computation (FNC) | Computational accuracy Unit consistency | Accuracy | 581 | 651 |
| Q&A | Financial Knowledge QA (FQA) | Question comprehension Knowledge coverage Answer accuracy | Judge Score | 990 | 22 |
| | Financial Data Description (FDD) | Trend accuracy Data consistency | Judge Score | 1461 | 311 |
| Prediction recognition | Emotion Recognition (ER) | Emotion classification accuracy Implicit information extraction | Accuracy | 600 | 2179 |
| | Stock Price Prediction (SP) | Trend judgment, Causal reasoning | Accuracy | 497 | 4498 |
| Information extraction | Financial Named Entity Recognition (FNER) | Recognition accuracy Entity classification correctness | Accuracy | 435 | 533 |

company news, sentiment-opposing articles (e.g., negative news during a stock rally), or temporally misaligned events. This design ensures that correct answers require genuine financial reasoning rather than superficial keyword matching.

Once queries and contexts were constructed, they were paired with task-specific prompts and submitted to a large language model (e.g., GPT-4o) to generate candidate answers. These answers were not directly included; instead, each data point underwent rigorous human annotation and validation. Specifically, every entry was independently reviewed by three senior financial experts with over five years of professional experience in equity research, investment analysis, or portfolio management at leading financial institutions (e.g., securities firms, asset management companies, banks). Experts evaluated both the correctness of the generated answers and the appropriateness of the assigned task category.

A data point was accepted into the final dataset only after full consensus among all three experts on answer validity, contextual consistency, and category correctness. In cases of disagreement, iterative reviews were conducted until unanimous agreement was reached. This multi-stage annotation protocol ensures that the dataset is factually reliable and faithfully reflects the standards of real-world financial reasoning and practice.

### 3.2 STATISTICS

The BizFinBench benchmark consists of a total of 7,605 entries, encompassing a wide variety of tasks designed to assess model performance across diverse financial challenges. By testing models on these tasks, we aim to evaluate not only their individual capabilities but also their ability to generalize across multiple facets of financial data analysis.

Table 2 provides a detailed breakdown of the dataset, including the evaluation dimensions, corresponding metrics, the number of instances per task, and the average token length per entry [3]. The dataset exhibits significant variability in input length, ranging from just 5 tokens to as many as 102,243 tokens. This broad range reflects the complexity and heterogeneity of real-world financial scenarios and presents a meaningful challenge for models to demonstrate their ability to process both short and long financial texts effectively.

### 3.3 KNOWLEDGE DEPTH AND BREADTH ANALYSIS

To further assess the dataset's cognitive complexity and domain coverage, we conducted additional analyses.

**Knowledge Depth.** We evaluate the reasoning depth of each instance using Bloom's Revised Taxonomy Sun et al. (2024), which defines six cognitive levels: *Remember (R)*, *Understand (U)*, *Apply (A)*,

---

[3]Detailed dataset information is provided in Appendix F.

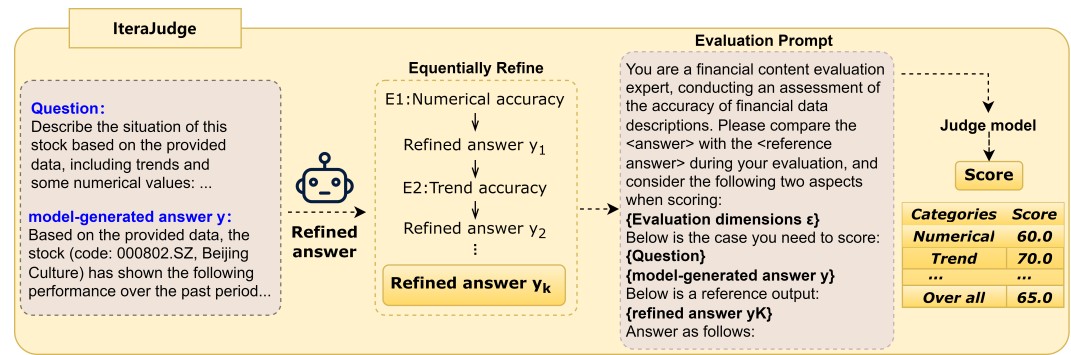

Figure 4: IteraJudge Pipeline.

*Analyze (An)*, *Evaluate (E)*, and *Create (C)*. Following an initial classification by DeepSeek-V3 and subsequent human verification, over 98% of the samples demand cognitive functions beyond simple memorization, with more than 93% requiring analytical reasoning. These findings demonstrate that BizFinBench extends well beyond factual retrieval, imposing substantial requirements on reasoning depth. The full distribution of cognitive levels is provided in Table 6 in the Appendix.

**Knowledge Breadth.** To assess domain coverage, we reference established qualification standards (e.g., CFA, FRM, Chinese Securities Exams) and define 21 financial subdomains. Each query can belong to multiple subdomains. The tagging process combines automated labeling with expert verification. BizFinBench spans a broad spectrum of financial domains, including accounting, valuation, risk management, compliance, and derivatives. These subdomains align closely with real-world financial tasks and professional certification standards. Detailed subdomain coverage statistics are reported in Table 7 in the Appendix.

### 3.4 ITERAJUDGE: AN INCREMENTAL MULTI-DIMENSIONAL EVALUATION FRAMEWORK

As shown in Figure 4, **IteraJudge** evaluation framework performs dimension-decoupled assessment through a three-phase pipeline:

1. Given question $q$ and initial answer $y \sim p_{\text{model}}(\cdot|q)$, we sequentially refine the output across dimensions $\mathcal{E} = \{e_1, \ldots, e_K\}$ via prompted LLM transformations:

$$y_k = \text{LLM}_{\text{refine}}(y_{k-1} \parallel \mathcal{P}(e_k, q)), \quad \text{where } y_0 = y \tag{1}$$

creating an interpretable improvement trajectory $\{y_k\}_{k=0}^{K}$.
2. The fully refined $y_K$ serves as an auto - generated quality benchmark.
3. A judge model computes the final score through contrastive evaluation:

$$\text{score}(y) = \text{LLM}_{\text{judge}}(q, y, y_K, \mathcal{E}) \tag{2}$$

where the delta $(y_K - y)$ quantitatively reveals the dimensional deficiencies of LLMs.

This *question-anchored*, iterative refinement process enables granular diagnosis while maintaining contextual consistency through explicit $q$-preservation in all steps.

## 4 EXPERIMENTS

This section summarizes the evaluated models (Section 4.1), including SOTA LLMs, inference-optimized models, and multimodal large language models. Section 4.2 describes the experimental setup. Section 4.3 presents key results across financial tasks, followed by the performance analysis of IteraJudge in Section 4.4.

## 4.1 EVALUATED MODELS

We conducted a systematic evaluation of current mainstream LLMs on BizFinBench. For closed-source models, we selected eight industry-recognized SOTA models: OpenAI's o3, 4o, GPT-5, GPT-5 mini and o4-mini, Google's Gemini-2.0-Flash, Gemini-2.5-Pro, and Anthropic's Claude-3.5-Sonnet. For open-source models, our evaluation covered both general-purpose LLMs including the Qwen2.5/3 series, Llama-3.1 series and Llama-4-Scout, as well as the financial-specialized Xuanyuan3-70B model. To comprehensively assess model capability boundaries, we also incorporated the DeepSeek-R1 series (including the R1-distill variant) which excels at complex reasoning tasks, the newly open-sourced reasoning model QwQ-32B and the recently released Qwen3 series with hybrid reasoning capabilities. Furthermore, to evaluate MLLMs on our benchmark, we extended our experiments to assess the performance of the Qwen-VL series of MLLMs[4].

Table 3: Performance Comparison of Large Language Models on BizFinBench. The models are evaluated across multiple tasks, with results color-coded to represent the top three performers for each task: golden indicates the top-performing model, silver represents the second-best result, and bronze denotes the third-best performance.

| Model | AEA | FNC | FTR | FTU | FQA | FDD | ER | SP | FNER | Average |
|---|---|---|---|---|---|---|---|---|---|---|
| Propretary LLMs | | | | | | | | | | |
| OpenAI o3 | 29.93 | 61.30 | 75.36 | 89.15 | 91.25 | 98.55 | 44.48 | 53.27 | 65.13 | 67.60 |
| OpenAI o4-mini | 19.15 | 60.10 | 71.23 | 74.40 | 90.27 | 95.73 | 47.67 | 52.32 | 64.24 | 63.90 |
| GPT-4o | 26.91 | 56.51 | 76.20 | 82.37 | 87.79 | 98.84 | 45.33 | 54.33 | 65.37 | 65.96 |
| GPT-5 mini | 15.36 | 88.11 | 84.05 | 91.14 | 87.80 | 97.92 | 23.27 | 55.13 | 60.02 | 66.98 |
| GPT-5 | 27.95 | 91.75 | 84.75 | 93.15 | 88.64 | 99.12 | 26.00 | 51.50 | 72.91 | 70.64 |
| Gemini-2.0-Flash | 37.38 | 62.67 | 73.97 | 82.55 | 90.29 | 98.62 | 22.17 | 56.14 | 54.43 | 64.25 |
| Gemini-2.5-pro | 40.18 | 87.57 | 87.74 | 87.25 | 87.56 | 96.35 | 43.15 | 53.12 | 75.71 | 73.18 |
| Claude-3.5-Sonnet | 27.84 | 63.18 | 42.81 | 88.05 | 87.35 | 96.85 | 16.67 | 47.60 | 63.09 | 59.27 |
| Open source LLMs | | | | | | | | | | |
| Qwen2.5-7B-Instruct | 12.68 | 32.88 | 39.38 | 79.03 | 83.34 | 78.93 | 37.50 | 51.91 | 30.31 | 49.52 |
| Qwen2.5-72B-Instruct | 28.98 | 54.28 | 70.72 | 85.29 | 87.79 | 97.43 | 35.33 | 55.13 | 54.02 | 64.41 |
| Qwen2.5-VL-3B | 2.62 | 15.92 | 17.29 | 8.95 | 81.60 | 59.44 | 39.50 | 52.49 | 21.57 | 32.01 |
| Qwen2.5-VL-7B | 8.87 | 32.71 | 40.24 | 77.85 | 83.94 | 77.41 | 38.83 | 51.91 | 33.40 | 48.86 |
| Qwen2.5-VL-32B | 26.42 | 50.06 | 62.16 | 83.57 | 85.30 | 95.95 | 40.50 | 54.93 | 68.36 | 63.21 |
| Qwen2.5-VL-72B | 32.11 | 54.11 | 69.86 | 85.18 | 87.37 | 97.34 | 35.00 | 54.94 | 54.41 | 64.46 |
| Qwen3-1.7B | 13.01 | 35.80 | 33.40 | 75.82 | 73.81 | 78.62 | 22.40 | 48.53 | 11.23 | 44.45 |
| Qwen3-4B | 21.97 | 47.40 | 50.00 | 78.19 | 82.24 | 80.16 | 42.20 | 50.51 | 25.19 | 53.04 |
| Qwen3-14B | 32.75 | 58.20 | 65.80 | 82.19 | 84.12 | 92.91 | 33.00 | 52.31 | 50.70 | 61.71 |
| Qwen3-32B | 29.85 | 59.60 | 64.60 | 85.12 | 85.43 | 95.37 | 39.00 | 52.26 | 49.19 | 63.19 |
| Qwen3-235B-A22B-Thinking | 28.13 | 87.27 | 84.23 | 88.69 | 91.76 | 99.34 | 32.67 | 52.56 | 52.15 | 68.27 |
| Qwen3-235B-A22B-Instruct | 43.22 | 82.52 | 82.88 | 92.46 | 89.63 | 99.67 | 28.83 | 56.74 | 48.37 | 69.72 |
| Xuanyuan3-70B | 15.16 | 19.69 | 15.41 | 80.89 | 86.51 | 83.90 | 29.83 | 52.62 | 37.33 | 46.82 |
| Llama-3.1-8B-Instruct | 3.93 | 22.09 | 2.91 | 77.42 | 76.18 | 69.09 | 29.00 | 54.21 | 36.56 | 40.76 |
| Llama-3.1-70B-Instruct | 17.96 | 34.25 | 56.34 | 80.64 | 79.97 | 86.90 | 33.33 | 52.16 | 45.95 | 55.28 |
| Llama 4 Scout | 22.49 | 45.80 | 44.20 | 85.02 | 85.21 | 92.32 | 25.60 | 55.76 | 43.00 | 55.49 |
| DeepSeek-V3 (671B) | 31.29 | 61.82 | 72.60 | 86.54 | 91.07 | 98.11 | 32.67 | 55.73 | 71.24 | 66.79 |
| DeepSeek-V3.1 | 37.22 | 91.04 | 84.16 | 90.78 | 90.65 | 99.32 | 25.63 | 52.15 | 54.75 | 69.52 |
| DeepSeek-R1 (671B) | 35.41 | 64.04 | 75.00 | 81.96 | 91.44 | 98.41 | 39.67 | 55.13 | 71.46 | 68.25 |
| QwQ-32B | 28.75 | 53.91 | 64.90 | 84.81 | 89.60 | 94.20 | 34.50 | 56.68 | 30.27 | 59.74 |
| DeepSeek-R1-Distill-Qwen-14B | 20.77 | 44.35 | 16.95 | 81.96 | 85.52 | 92.81 | 39.50 | 50.20 | 52.76 | 55.06 |
| DeepSeek-R1-Distill-Qwen-32B | 27.42 | 51.20 | 50.86 | 83.27 | 87.54 | 97.81 | 41.50 | 53.92 | 56.80 | 62.15 |

## 4.2 EXPERIMENT SETTING

All LLMs were configured with a maximum generation length of 1,024 tokens, temperature parameter $T = 0$, and batch size $B = 1000$. We employed GPT-4o as the unified evaluation judge. Open-source models were deployed on an 8×NVIDIA H100 cluster, while closed-source models were accessed via their official APIs. The complete evaluation required approximately 10 hours with a total computational cost of $21,000.

---

[4]The specific details of the relevant models can be found in the Appendix D.

To ensure standardized outputs and enable automated evaluation, all models were required to produce strictly JSON-formatted responses with two mandatory fields: ① *Chain-of-Thought (CoT)* — a detailed reasoning trace with intermediate steps; ② *Answer* — the final conclusion. [5]

Comprehensive details regarding output formatting and the automated evaluation protocol, including the JSONL requirements, are provided in Appendix C.

### 4.3 MAIN RESULTS

Our evaluation on the BizFinBench benchmark reveals distinct capabilities of LLMs in the financial domain. All results as shown in Table 3. In AEA, Qwen3-235B-A22B-Instruct leads with 43.22, followed by Gemini-2.5-pro (40.18) and Gemini-2.0-Flash (37.38). For knowledge-intensive tasks, results are split: GPT-5 ranks first in FNC (91.75) and FTU (93.15), Gemini-2.5-pro in FTR (87.74), Qwen3-235B-A22B-Thinking in FQA (91.76), and Qwen3-235B-A22B-Instruct in FDD (99.67). In ER, top proprietary models score around 45–48, while the best open-source reaches 41.50. For FNER, Gemini-2.5-pro (75.71) and GPT-5 (72.91) lead, with DeepSeek-R1 (71.46) and DeepSeek-V3 (71.24) also competitive. On average, Gemini-2.5-pro ranks first (73.18), ahead of GPT-5 (70.64) and Qwen3-235B-A22B-Instruct (69.72).

Three insights emerge. First, scaling boosts performance: within Qwen3, averages rise from 44.45 (1.7B) to 63.19 (32B). Second, AEA is the hardest task, with scores from 2.62 (Qwen2.5-VL-3B) to 43.22 (Qwen3-235B-A22B-Instruct). Third, structured tasks like FDD exceed 95 across models, but ER stays below 50, even for top models. Two further observations: distilled models such as DeepSeek-R1-Distill-Qwen-32B retain strong FTU ability (83.27) but weaken in FTR (50.86), and SP results cluster tightly near 52–56, hinting at a performance ceiling without domain-specific signals.

### 4.4 ITERAJUDGE ABLATION EXPERIMENTS

To rigorously validate the effectiveness of IteraJudge, we conducted ablation experiments on the FDD and FTU benchmark datasets. We selected Qwen2.5-7B-Instruct as the evaluated model and employed GPT-4o, DeepSeek-V3, Gemini-2.0-Flash, and Qwen2.5-72B-Instruct as judge models to evaluate its generated responses. Three sets of experiments were designed: (1) expert evaluation, (2) the vanilla LLM-as-a-Judge approach, and (3) the full IteraJudge framework. We take the Spearman correlation between the evaluation methods, i.e., vanilla LLM-as-a-Judge and IteraJudge.

Detailed experimental results are provided in Table 8 in the Appendix. In summary, compared to the vanilla LLM-as-a-Judge approach, IteraJudge achieves a maximum improvement of 17.24% and a minimum improvement of 3.09% on the FDD benchmark dataset. On the FTU benchmark dataset, it shows a maximum improvement of 11.37% and a minimum improvement of 4.44%. These results confirm the effectiveness of IteraJudge in mitigating evaluation bias.

### 5 CONCLUSION

In this work, we propose BizFinBench, i.e., the first open-source Chinese benchmark dataset, which consists of the dataset deeply integrated with real-world financial business scenarios and a iterative calibration-based evaluation framework, i.e., IteraJudge. We conducted a comprehensive evaluation of 30 SOTA LLMs, encompassing both closed-source and open-source models, across multiple task dimensions. Our results reveal significant performance gaps between existing LLMs and human-level expectations in several business-critical areas, highlighting the unique challenges of financial artificial intelligence. We find that no model dominates every task, while OpenAI o4-mini, GPT-5, Gemini-2.5-Pro, Qwen3-235B-A22B-Thinking, and Qwen3-235B-A22B-Instruct corresponds to the best performance in diverse metrics. In addition, experimental results also demonstrate that closed-source models place in the top three on eight of nine subtasks. Furthermore, extensive experimental results reveal significant advantages of IteraJudge. BizFinBench serves not only as a rigorous benchmark for evaluating financial reasoning capabilities, but also as a practical guide for deploying LLMs in real-world financial applications. We believe this benchmark can accelerate progress in the development of trustworthy, high-performing financial language models.

---

[5]Formatting examples for each dataset are provided in Appendix G, and evaluation prompts are listed in Appendix E.

ETHICS STATEMENT

This work introduces a Chinese benchmark in the financial domain, built from real-world user queries. To protect user privacy, all potentially sensitive or personally identifiable information (e.g., names, account details, or contact information) has been thoroughly anonymized or removed. The released dataset contains only de-identified text and does not include confidential financial records.

The benchmark is intended strictly for research purposes, such as advancing natural language processing and evaluation in financial applications. It should not be directly applied to production systems for financial decision-making without rigorous validation and additional safeguards. We have followed ethical standards for data collection, anonymization, and release, and have carefully considered potential risks of bias, misuse, or unintended societal impact.

REPRODUCIBILITY STATEMENT

To ensure the reproducibility of our results, we have made substantial efforts to provide all necessary details and materials. Specifically, Section 3.1 presents the complete process of dataset construction, including data collection strategies. Furthermore, the benchmark setup and evaluation procedures are thoroughly described in Section 4. All evaluation metrics are clearly defined to facilitate independent verification and replication of our experiments by the research community.

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

## A USE OF LLMs

Large language models (LLMs) were employed in multiple stages of this work. During dataset construction, LLMs were used to assist in data cleaning and annotation, while ensuring that all outputs were manually reviewed to prevent privacy leakage and annotation errors. For evaluation, we adopted LLMs as judges to provide comparative assessments of model outputs in financial tasks. This design follows recent practices in benchmarking and allows for scalable, consistent evaluation.

We acknowledge that LLM-based evaluation may inherit biases or inaccuracies from the underlying models. To mitigate these risks, we combined automated judgments with careful sampling-based human validation. Overall, the use of LLMs was limited to auxiliary roles, and all critical results were cross-checked to ensure reliability.

## B    LIMITATIONS

In this work, we propose a novel benchmark and conduct a comprehensive analysis of different LLMs' capabilities in solving financial business problems. However, several limitations remain:

(1) Our method for extracting final answers from model outputs is not yet perfect. In some cases, this method fails to locate an answer, leading to reported accuracy being an approximate lower bound. Additionally, due to potential formatting differences between the extracted answers and the ground truth, we employ a rule-based approach to measure exact matches between the two, which may introduce an estimated 2% error in our experiments.

(2) Our benchmark is primarily based on currently available financial data and task settings. Although it covers multiple key sub-tasks, some business scenarios may still be underrepresented. For example, highly specialized financial tasks such as complex derivatives pricing, risk management modeling, or decision support based on real-time market data are not yet fully reflected in our benchmark. This implies that our evaluation results may not completely capture LLMs' real-world performance in more complex financial scenarios.

(3) While we evaluate multiple SOTA LLMs under the same computational environment to ensure fairness, model performance may still be influenced by training data, inference strategies, and hyperparameter settings. Additionally, discrepancies between inference mechanisms in API-based and locally deployed models could introduce experimental biases.

(4) Our evaluation primarily focuses on models' abilities in single-turn question answering and task completion. However, in real-world applications, financial decision-making is often a complex, multi-step process involving long-term reasoning, external tool utilization, and multi-turn interactions. The current evaluation framework does not fully cover these aspects, highlighting the need for further expansion to better reflect LLMs' potential applications in financial business scenarios.

For future work, we plan to optimize the answer extraction method to enhance evaluation accuracy and explore more advanced metrics to mitigate errors caused by format mismatches. Additionally, we aim to expand the benchmark's coverage by incorporating more challenging financial tasks and refining experimental settings to improve reproducibility and fairness.

## C    JSONL OUTPUT AND EVALUATION PROTOCOL

During evaluation, outputs were first validated against the JSONL schema. Conforming responses were parsed and scored directly. As shown in Figure 5, certain cases failed due to minor formatting errors. To address this, a two-stage strategy was adopted: if strict parsing failed, an auxiliary LLM performed semantic analysis on the raw output to recover the intended answer. This ensures that formatting mistakes do not unfairly penalize model performance.

## D    MODEL DETAIL

To better ensure the comprehensiveness and robustness of our evaluation, we selected a wide range of models that differ in architecture, parameter size, training objectives, and domain specialization. Table 4 presents detailed information on the 30 evaluation models used in this study.

## E    INSTRUCTION

Figure 6, Figure 7, and Figure 8 illustrate the instructions used for model evaluation on the open-ended answer dataset.

## F    DATASET DETAILS

The details of each dataset type are as follows.

- Anomalous Event Attribution (AEA): This dataset evaluates a model's ability to trace financial anomalies based on comprehensive textual inputs, including news articles, financial

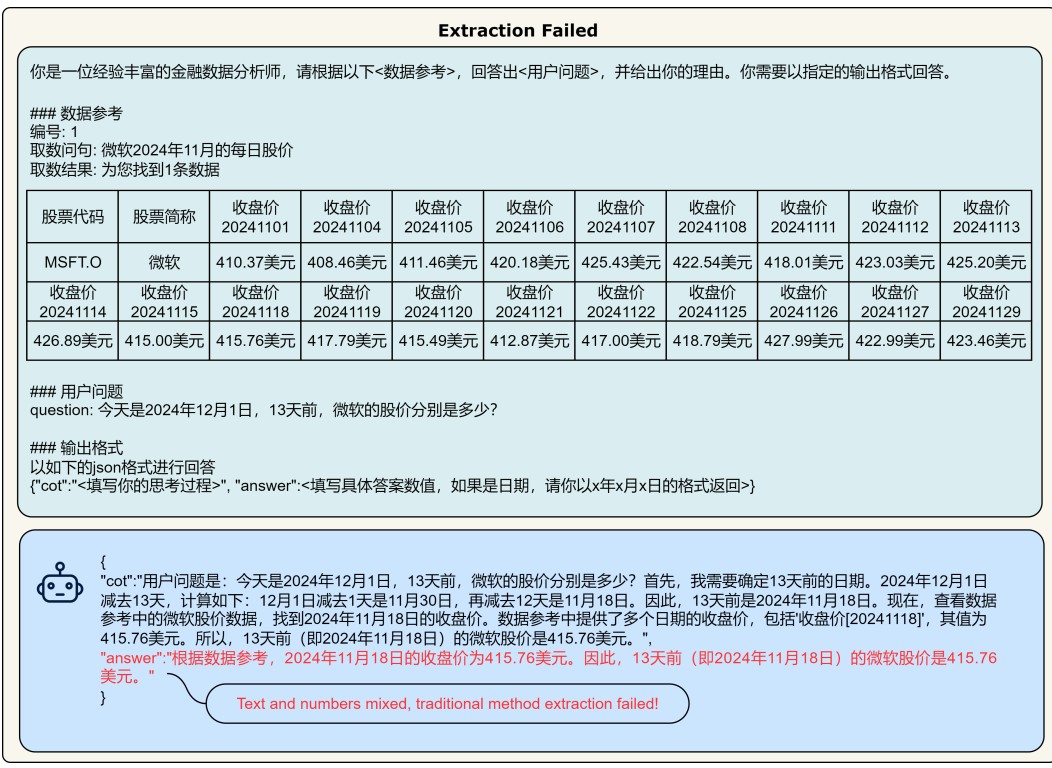

Figure 5: Example of a JSONL case where the model output does not conform to the required format, resulting in the evaluation being marked as incorrect.

Table 4: Summary of Large Language Models Evaluated on BizFinBench. * indicates that the model is a Mixture-of-Experts (MoE) model.

| Model | Size | Open source | Evalation | Release date | Domain |
|---|---|---|---|---|---|
| GPT-4o OpenAI (2023) | – | ✗ | API | 01/29/2025 | General |
| OpenAI o3 | – | ✗ | API | 04/16/2025 | General |
| OpenAI o4-mini | – | ✗ | API | 04/16/2025 | General |
| GPT-5 mini | – | ✗ | API | 08/08/2025 | General |
| GPT-5 | – | ✗ | API | 08/08/2025 | General |
| Gemini-2.0-Flash Team et al. (2023) | – | ✗ | API | 12/11/2024 | General |
| Gemini-2.5-pro | – | ✗ | API | 06/18/2025 | General |
| Claude-3.5-Sonnet Claude (2024) | – | ✗ | API | 06/20/2024 | General |
| Qwen2.5-Instruct Bai et al. (2023) | 7B,72B | ✓ | Local | 09/19/2024 | General |
| Qwen2.5-VL Bai et al. (2025) | 3B,7B,32B,72B | ✓ | Local | 01/28/2025 | General |
| Qwen3 Yang et al. (2025) | 1.7B,4B,14B,32B,235B* | ✓ | Local | 04/29/2025 | General |
| XuanYuan3-70B-Chat Team (2024) | 70B | ✓ | Local | 09/06/2024 | Finance |
| Llama-3.1-Instruct Dubey et al. (2024) | 8B,70B | ✓ | Local | 07/24/2024 | General |
| Llama 4 | 109B* | ✓ | Local | 04/05/2025 | General |
| DeepSeek-V3 Liu et al. (2024) | 671B* | ✓ | Local | 12/26/2024 | General |
| DeepSeek-V3.1 Liu et al. (2024) | 671B* | ✓ | Local | 08/21/2025 | General |
| DeepSeek-R1 Liu et al. (2024) | 671B* | ✓ | Local | 12/26/2024 | General |
| QwQ-32B Team (2025) | 32B | ✓ | Local | 03/06/2025 | General |
| DeepSeek-R1-Distill-Qwen Liu et al. (2024) | 14B,32B | ✓ | Local | 12/26/2024 | General |

你是一个金融内容评测专家，正在进行金融数据描述准确性的评估。

你的打分需要考虑两个方面：

1. 数据错用：<answer>中的指标数字应该和<question>中的对应上，不应该出现指标错用、时间错用等情况，

   例如：从55.32增长到59.14描述成从55.24增长到58.32。

2. 数据描述：只需判断<answer>中是否存在描述与具体数据相背的情况，如果有则得0分。

   例如：一连串数值越来越大，描述却是递减、两两比较错误，或最大、最小值判断错误、涨跌幅大于零说成下跌、

   主力资金小于零说成资金流入。当<question>中未取到数或取到的数据为空时，<answer>中回答不能说该数据为0，如果有则得0分。

| 分数 | 描述 |
|------|------|
| 100 | 完全正确。趋势描述和数据描述的均完全正确，且语言流程，无幻觉。 |
| 60 | 部分错误。数据趋势描述正确，但数据值描述错误，例如从55.32增长到59.14描述成从55.24增长到58.32。 |
| 0 | 错误较多。数据趋势描述错误即不得分，例如数据趋势是越来越大，描述是递减。 |

以下是你需要评分的案例：

<question>

[question]

</question>

<answer>

[predict_result]

</answer>

要求：

返回结果以json格式展示，参考： {"评分分数":"xx","描述": "xxxx"}

回答如下：

Figure 6: The instructions utilized in the evaluation of the FDD dataset.

reports, timestamped events, and structured data such as tables and market statistics—all presented in text form. The task requires identifying causal relationships underlying sudden market fluctuations while filtering out extensive irrelevant or misleading information. In the most challenging configuration, up to 20 distractor documents are introduced, each containing long, semantically plausible, and carefully crafted text that mimics real-world noise—such as unrelated corporate announcements, temporally close but irrelevant events, or spurious financial correlations. These distractors are designed to be highly deceptive, increasing the difficulty of isolating genuine signals. The dataset thus provides a rigorous evaluation of a model's reasoning, contextual filtering, and long-range dependency tracking capabilities in complex, document-rich environments.

- Financial Numerical Computation (FNC): This dataset assesses the model's ability to perform accurate numerical calculations in financial scenarios, including interest rate calculations, return on investment (ROI), and financial ratios.

- Financial Time Reasoning (FTR): This dataset tests the model's ability to understand and reason about time-based financial events, such as predicting interest accruals, identifying the impact of quarterly reports, and assessing financial trends over different periods.

- Financial Tool Usage (FTU): This dataset evaluates the model's ability to comprehend user queries and effectively use financial tools to solve real-world problems. It covers scenarios like investment analysis, market research, and information retrieval, requiring the model to select appropriate tools, input parameters accurately, and coordinate multiple tools when needed.

- Financial Knowledge QA (FQA): This dataset evaluates the model's understanding and response capabilities regarding core knowledge in the financial domain. It spans a wide range of financial topics, encompassing key areas such as fundamental financial concepts, financial markets, investment theory, macroeconomics, and finance.

- Financial Data Description (FDD): This dataset measures the model's ability to analyze and describe structured and unstructured financial data, such as balance sheets, stock reports, and financial statements.

你是一个金融内容评测专家，可以通过以下几个维度对模型生成得内容进行合理、正确的评分；

1. 准确性 (Accuracy)

评估答案是否完全符合金融领域的标准知识和常识，重点关注概念、公式、计算、结论是否正确。

| 等级 | 描述 | 分数范围 |
|---|---|---|
| 5分 | 完全正确。所有概念、公式、计算、结论无误，完全符合金融理论和实际应用。 | 90-100 |
| 4分 | 基本正确。核心概念和计算正确，少量细节或边缘部分略有偏差。 | 75-89 |
| 3分 | 部分错误。关键概念正确，但存在明显错误或不一致，部分计算或结论偏差较大。 | 60-74 |
| 2分 | 错误较多。多个关键部分出错，影响了整体理解和计算过程。 | 40-59 |
| 1分 | 严重错误。答案完全错误，核心理论或计算无法支持结论。 | 0-39 |

...

总评分：

- 每个维度的最高分为 25分，总分 100分。
- 可以根据实际需求调整维度的权重。常见的权重分配如下：
  - 准确性：40%
  - 完整性：30%
  - 分析深度：20%
  - 清晰度：10%

总评分示例：

- 准确性： 23分（40%）
- 完整性： 21分（30%）
- 分析深度： 17分（20%）
- 清晰度： 8分（10%）
总分 = 23×0.40+21×0.30+17×0.20+8×0.10=9.2+6.3+3.4+0.8=19.723

以下是你需要评分的案例：
<question>
**[question]**
</question>

<answer>
**[predict_result]**
</answer>

要求：
返回结果以json格式展示，参考：
{"评分结果":{"准确性":{"得分":23,"描述":"该答案准确地解释了货币政策和财政政策的作用及其常见措施，如降息、量化宽松、政府支出、减税等，均符合金融领域的标准知识和实际应用。概念和措施没有明显错误，符合经济学和金融学的基本原理。"},"完整性":{"得分":22,"描述":"答案完整地涵盖了货币政策和财政政策的关键措施，并指出了这两者结合使用的可能性。提到了实施政策时的时机、潜在副作用以及国际协调的需要，涉及了多个重要方面。不过，某些细节，如具体的政策效果评估和可能的长期影响，未做更深入探讨。"},"分析深度":{"得分":18,"描述":"从多个维度分析了货币政策和财政政策的作用，并提到了一些重要的影响因素，如时机、副作用和国际合作等。尽管涉及到了一些复杂因素，但在某些方面（如副作用的深度分析、政策效果的具体评估）可进一步加强讨论。"},"清晰度":{"得分":24,"描述":"表达非常清晰，逻辑严密，语言流畅。各部分内容结构合理，易于理解，专业术语准确无误。整体条理清晰，读者能轻松跟随作者的思路。"}},"总评分":{"准确性":9.2,"完整性":6.6,"分析深度":3.6,"清晰度":2.4,"最终评分":21.8,"总分":87.2}}

你的答案如下：

Figure 7: The instructions utilized in the evaluation of the FQA dataset.

你是一个专业的规划智能体评测模型，负责对其他模型使用规划工具的能力进行评估。
被评测模型可使用的工具列表：
[apis_information]

用户问题及历史对话：
[question]

被评测模型的输出：
[predict_result]

评估标准（总分100分）：
1. 准确性 (50分)：
 - 工具选择的合理性 (30分)：评估所选工具是否适合解决用户的问题
 - 输入参数的合理性 (20分)：检查每个工具的输入参数是否符合对应工具输入参数的要求。
2. 实用性 (50分)：
 - 解决方案的有效性 (50分)：评估提供的解决方案是否能够有效解决问题或达成目标。

评分体系：
被评测模型将根据上述标准获得一个介于1至100分之间的综合评分，分数越高表示性能越优秀。
当且仅当被评测模型的表现完全符合标准时，才会得到满分100分。
请在评估过程中保持客观公正，避免任何形式的偏见。

输出格式要求：
评价需包含详细的解释以及最终得分，严格按照如下格式呈现："[[分数]]"。

示例输出格式：
<开始输出>
评价证据：这里填写您对评测结果的具体分析和理由，不超过100字。
评分：[[实际分数]]
<输出结束>

现在，请基于以上指导原则开始您的评估工作。

Figure 8: The instructions utilized in the evaluation of the FTU dataset.

- Emotion Recognition (ER): This dataset evaluates the model's capability to recognize nuanced user emotions in complex financial market environments. The input data encompasses multiple dimensions, including market conditions, news articles, research reports, user portfolio information, and queries. The dataset covers six distinct emotional categories: optimism, anxiety, negativity, excitement, calmness, and regret.

- Stock Price Prediction (SP): This dataset evaluates the model's ability to predict future stock prices based on historical trends, financial indicators, and market news.

- Financial Named Entity Recognition (FNER): This dataset focuses on evaluating the model's ability to identify and classify financial entities such as company names, stock symbols, financial instruments, regulatory agencies, and economic indicators.

Table 5 presents their maximum token length, minimum token length, and average length.

Table 5: Financial Datasets Query Token Length Statistics. This table presents token length statistics for queries in financial datasets, including minimum (Min), maximum (Max), average (Avg) token counts, and total query count (Count).

| Dataset | Min | Max | Avg | Count |
|---|---|---|---|---|
| NER | 415 | 1,194 | 533.1 | 433 |
| FTU | 4,169 | 6,289 | 4,555.5 | 641 |
| AEA | 4,472 | 102,243 | 28,357 | 1,888 |
| ER | 1,919 | 2,569 | 2,178.5 | 600 |
| FNC | 287 | 2,698 | 650.5 | 581 |
| FDD | 26 | 645 | 310.9 | 1,461 |
| FTR | 203 | 8,265 | 1,162.0 | 514 |
| FQA | 5 | 45 | 21.7 | 990 |
| SP | 1,254 | 5,532 | 4,498.1 | 497 |

# G  DATASET EXAMPLE

# H  ADDITIONAL TABLES FOR KNOWLEDGE ANALYSIS

Table 6: Distribution of cognitive complexity levels in BizFinBench based on Bloom's Revised Taxonomy.

| Cognitive Level | Count | Proportion |
|---|---|---|
| Remember (R) | 142 | 1.9% |
| Understand (U) | 320 | 4.2% |
| Apply (A) | 1,518 | 20.0% |
| Analyze (An) | 4,940 | 65.0% |
| Evaluate (E) | 633 | 8.3% |
| Create (C) | 52 | 0.7% |

# I  ADDITIONAL EXPERIMENTAL RESULTS

# J  OTHER

Figure 9: An example instance from the Financial Numerical Computation dataset, in which the data table is formatted using Markdown syntax.

---

**Financial Time Reasoning**

你是一位经验丰富的金融数据分析师，请根据以下<数据参考>，回答出<用户问题>，并给出你的理由。你需要以指定的输出格式回答

### 数据参考
编号: 1
取数问句: 平安银行2023年10月1日前后10日的收盘价明细
取数结果: 为您找到1条数据

| 股票代码 | 股票简称 | 收盘价 20230921 | 收盘价 20230922 | 收盘价 20230925 | 收盘价 20230926 | 收盘价 20230927 | 收盘价 20230928 | 收盘价 20231009 | 收盘价 20231010 | 收盘价 20231011 | 收盘价 20231011 |
|---|---|---|---|---|---|---|---|---|---|---|---|
| 000001.SZ | 平安银行 | 10.09元 | 10.28元 | 10.26元 | 10.20元 | 10.21元 | 10.23元 | 10.14元 | 10.05元 | 10.02元 | 10.11元 |

编号: 2
取数问句: 平安银行2024年10月1日前后10日的收盘价明细
取数结果: 为您找到1条数据

| 股票代码 | 股票简称 | 收盘价 20240923 | 收盘价 20240924 | 收盘价 20240925 | 收盘价 20240926 | 收盘价 20240927 | 收盘价 20240930 | 收盘价 20241008 | 收盘价 20241009 | 收盘价 20241010 | 收盘价 20241011 |
|---|---|---|---|---|---|---|---|---|---|---|---|
| 000001.SZ | 平安银行 | 9.79元 | 10.12元 | 10.25元 | 10.90元 | 11.17元 | 11.96元 | 12.63元 | 11.43元 | 11.98元 | 11.72元 |

编号: 3
取数问句: 平安银行2022年10月1日前后10日的收盘价明细
取数结果: 为您找到1条数据

| 股票代码 | 股票简称 | 收盘价 20220921 | 收盘价 20220922 | 收盘价 20220923 | 收盘价 20220926 | 收盘价 20220927 | 收盘价 20220928 | 收盘价 20220929 | 收盘价 20220930 | 收盘价 20221010 | 收盘价 20221011 |
|---|---|---|---|---|---|---|---|---|---|---|---|
| 000001.SZ | 平安银行 | 11.18元 | 11.04元 | 11.04元 | 10.75元 | 10.90元 | 10.86元 | 10.61元 | 10.59元 | 10.22元 | 10.23元 |

### 用户问题
question: 今天是2024年11月15日，请问国庆节后平安银行股价首次突破12元的日期是哪一天？

### 输出格式
以如下的json格式进行回答
{"cot":"<填写你的思考过程>", "answer":<填写具体答案，如果是数值，请统一表示成小数，最多小数点后保留四位；如果是日期，请你以x年x月x日的格式返回>}

---

{
 "cot": "根据提供的数据，我们需要查看2024年10月1日前后10日的股价明细，以确定国庆节后平安银行股价首次突破12元的日期。从数据中可以看到，2024年10月8日的收盘价为12.63元，这是国庆节后首次突破12元的日期。",

 "answer": "2024年10月8日"
}

Figure 10: An example instance from the Financial Time Reasoning.

---

**Financial Knowledge QA**

什么是货币时间价值?

货币时间价值是指货币在不同时间点上的价值差异。简单来说，今天的1元钱比将来的1元钱更有价值。这是因为货币可以被投资以获得收益，或者由于通货膨胀，未来的货币购买力可能会下降。因此，在金融决策中，考虑货币的时间价值是非常重要的。

货币时间价值的概念主要基于以下几个方面:

1. 投资收益: 如果你现在有100元，你可以选择将它存入银行或投资于其他项目，从而在未来获得更多的钱。因此，今天的100元比未来的100元更有价值，因为你有机会通过投资增加它的价值。

2. 通货膨胀: 随着时间的推移，由于通货膨胀的影响，货币的购买力会逐渐下降。这意味着，同样数量的货币在未来可能无法购买到与现在相同数量的商品和服务。

3. 风险与不确定性: 未来是不确定的，承诺在未来支付的款项可能因为各种原因（如违约、经济衰退等）而无法兑现。因此，人们通常更偏好当前的确定性收益而非未来的不确定性收益。

在金融学中，货币时间价值通常通过现值（Present Value, PV）和终值（Future Value, FV）的概念来量化。现值是指未来某一时点的一定金额按照一定的利率折算到现在的价值；终值则是指现在的一笔资金按照一定的利率增长到未来某一时点的价值。通过这些计算，可以帮助个人和企业做出更合理的财务决策。

Figure 11: An example instance from the Financial Knowledge QA dataset.

**Financial Data Description**

根据下面的提供的数据描述这支股票的情况，包括变化趋势和一些数值：

| 股票代码 | 股票简称 | 涨跌幅：前复权 20250110 | 涨跌幅：前复权 20250113 | 涨跌幅：前复权 20250114 | 涨跌幅：前复权 20250115 | 涨跌幅：前复权 20250116 | 涨跌幅：前复权 20250117 |
|---|---|---|---|---|---|---|---|
| 000802.SZ | 平安银行 | -9.97% | -1.48% | 3.45% | -2.75% | 0.0% | 1.19% |

| 涨跌幅：前复权 20250120 | 涨跌幅：前复权 20250121 | 涨跌幅：前复权 20250122 | 涨跌幅：前复权 20250123 | 涨跌幅：前复权 20250124 | 涨跌幅：前复权 20250127 | 涨跌幅：前复权 20250205 | 涨跌幅：前复权 20250206 |
|---|---|---|---|---|---|---|---|
| 1.33% | 2.62% | -2.12% | -6.51% | -1.55% | 10.06% | -10.0% | -6.19% |

| 收盘价：不复权 20250206 | 开盘价：不复权 20250206 | 最高价：不复权 20250206 | 最低价：不复权 20250206 | | | | |
|---|---|---|---|---|---|---|---|
| 5.91元 | 5.64元 | 6.04元 | 5.67元 | | | | |

根据提供的数据，我们可以对股票代码为000802.SZ的北京文化股票在2025年1月10日至2025年2月6日这段时间内的表现进行如下描述：

### 涨跌幅变化趋势
1. **2025年1月10日**：股票大幅下跌9.97%，这可能是由于市场对该公司或行业出现了负面消息，导致投资者信心下降。
2. **2025年1月11日至1月14日**：股票继续下跌1.48%，但随后在1月14日反弹3.45%，显示出市场情绪有所恢复。
3. **2025年1月15日至1月16日**：股票再次下跌2.75%，但1月16日没有变化，市场进入观望状态。
4. **2025年1月17日至1月21日**：股票连续小幅上涨，涨幅分别为1.19%、1.33%和2.62%，显示出市场信心逐渐增强。
5. **2025年1月22日至1月24日**：股票再次出现波动，1月22日下跌2.12%，1月23日大幅下跌6.51%，1月24日小幅下跌1.55%。
6. **2025年1月27日**：股票大幅上涨10.06%，显示出市场情绪的强烈反弹。
7. **2025年2月5日至2月6日**：股票再次大幅下跌10.00%，随后在2月6日继续下跌6.19%，市场情绪再次转为悲观。

### 数值分析
- **2025年2月6日的收盘价**：5.91元
- **2025年2月6日的开盘价**：5.67元
- **2025年2月6日的最高价**：6.04元
- **2025年2月6日的最低价**：5.67元

### 总体趋势
- **波动性**：在这段时间内，北京文化的股票价格波动较大，多次出现大幅涨跌，显示出市场对该股票的不确定性较高。
- **市场情绪**：市场情绪在短期内多次变化，从大幅下跌到强烈反弹，再到再次下跌，显示出投资者对该公司或行业的看法存在较大分歧。
- **价格区间**：从2025年1月10日至2025年2月6日，股票价格在5.67元至6.04元之间波动，整体价格区间较窄。

### 结论
北京文化在这段时间内的股票表现较为不稳定，市场情绪波动较大。投资者应密切关注公司基本面和市场消息，以做出更合理的投资决策。

Figure 12: An example instance from the Financial Data Description.

Figure 13: An example instance from the Financial Named Entity Recognition.

Table 7: Subdomain coverage statistics of BizFinBench, aligned with professional certification standards.

| Subdomain | Samples | Subdomain | Samples |
|---|---|---|---|
| Financial Statement Extraction | 3,628 | Risk Propagation Reasoning | 549 |
| Market Trend Forecasting | 2,860 | Regulatory Compliance Reasoning | 95 |
| Financial Indicator Computation | 2,600 | Policy Impact Forecasting | 31 |
| Business Decision Reasoning | 1,835 | Valuation Model Calculation | 52 |
| Financial Analysis Reasoning | 1,178 | Risk Quantification | 118 |
| Market Announcement Extraction | 2,618 | Business Metric Forecasting | 84 |
| Textual Relation Extraction | 319 | Business Data Calculation | 259 |
| Regulatory Document Extraction | 48 | Financial Instruments | 168 |
| Risk Event Forecasting | 351 | Legal & Regulatory Clauses | 93 |
| Basic Concepts | 623 | Business Processes | 17 |
| Bond Duration | 3 | | |

Table 8: Comparative Evaluation of Judgment Methods Across Different LLM Judges

| Methods | Financial Data Description | Financial Tool Usage |
|---|---|---|
| LLM as a judge (GPT-4o) | Spearman: 0.4848 | Spearman: 0.8000 |
| Ours (GPT-4o) | Spearman: 0.5684 | Spearman: 0.8667 |
| LLM as a judge (DeepSeek-V3) | Spearman: 0.4685 | Spearman: 0.7500 |
| Ours (DeepSeek-V3) | Spearman: 0.4830 | Spearman: 0.7833 |
| LLM as a judge (Gemini-2.0-Flash) | Spearman: 0.3763 | Spearman: 0.7333 |
| Ours (Gemini-2.0-Flash) | Spearman: 0.4087 | Spearman: 0.8167 |
| LLM as a judge (Qwen2.5-72B-Instruct) | Spearman: 0.3112 | Spearman: 0.7000 |
| Ours (Qwen2.5-72B-Instruct) | Spearman: 0.4282 | Spearman: 0.7500 |

**Financial Tool Usage**

请根据用户的输入和历史对话信息回复用户。您可以选择调用外部工具来实现。你可以假设api的返回结果。下面是调用需求和可用api的信息。

调用需求：
1. 请在"Thought"中提供你的思考过程，包括用户意图分析，是否调用api，如何调用api。
2. 当需要通过调用API满足使用者的要求时，请使用以下格式提供所需的调用信息：
   - Action：要调用的API名称。
   - Action Input：调用API所需的参数信息，格式为JSON。
   例如：
   Action："api_name_A"
   Action Input：{"parameter_name_A.1"："parameter_value_A.1", ...}
   Action："api_name_B"
   Action Input：{"parameter_name_B.1"："parameter_value_B.2", ...}, ...
3. API之间可能存在交互关系，需要将上一次API调用返回的参数值作为下一次API调用的参数值。请使用
"previous_API_name.return_parameter_name"作为新API调用的参数值。
4. 如果用户的需求可以在不调用API的情况下得到满足，那么就不会进行任何API调用操作。输出格式如下：
Thought: [你的思考过程]
5. 确保所有API名称和参数名称与提供的API信息保持一致，参数值应从上下文中提取，不应虚构。
6. 注意当前日期：2025年1月19日，星期日。

以下是你可以使用的api及其参数列表：**[**
{
"name": "FinQuery",
"description": "金融数据查询工具，用于获取标的相关的金融数据，如宏观数据、财务数据、行情数据、交易数据、个人账户数据、自选股
等。支持股票、美股、港股、基金、指数、宏观、可转债、期货等。",
"required_parameters": [{"name": **"indicators"**, "type": "array", "description": "需要查询的金融指标列表（例如：['AAPL营收', '标普500市
盈率']）。注意：指标中不应包含时间信息，时间范围请通过`time_range`参数指定。"}],
"optional_parameters": [{"name": **"company_identifier"**,"type": "string","description": "公司名称或股票代码（例如：'Apple
Inc.' 或 'AAPL'）。在查询指标时，如果涉及公司名称或股票代码，则必须提供公司名称或股票代码。"},{"name": **"time_range"**,
"type": "string", "description": "查询的时间范围，支持多种格式：单个年份如'2022'、年份范围如'2022-2023'、单个季度如'2022Q1'、季度范
围如'2022Q1-2023Q4'、单个月份如'2022-01'、月份范围如'2022-01-2023-12'、单个日期'2022-01-01'、日期范围如'2022-01-01-2023-12-
31'、相对时间如'last 5 years', 'last 3 months', 'last 7 days'、默认时间范围如'last 1 year'。如果未提供该参数，则默认查询最近一年的数据。对
于高频指标（如行情数据、交易数据），建议使用 'last 7 days' 或 'last 1 month'。"}],
"return_info": [{"name": **"data"**,"type": "object","description": "请求的金融数据，以结构化格式返回。"}]
},
**...... ]**

输出格式：
{
"Thought": "你需要仔细阅读并分析历史对话和用户的输入，提供你的思考过程，包括用户意图分析，是否调用api，如何调用api",
"Actions": [{"Action": "要调用的API_A名称", "ActionInput": {"被调用的API_A的参数": "值"}},{"Action": "要调用的API_B名
称", "ActionInput": {"被调用的API_B的参数": "值"}}...]
}

下面是用户的输入和历史对话信息：
{'User': '中航电测的股东结构有什么变化吗？'}

请按照输出格式要求，输出你的Thought和Actions!

```
{
  "Thought": "用户询问中航电测的股东结构是否有变化。为了回答这个问题，我需要调用PublicCompanyInfo API来获取中航电测
  的股东信息，特别是最近的变化情况。",
  "Actions": [
    {
      "Action": "PublicCompanyInfo",
      "ActionInput": {
        "stock_code": "中航电测",
        "info_type": "股东信息",
        "time_range": "last 1 month"
      }
    }
  ]
}
```

Figure 14: An example instance from the Financial Tool Usage.

**Anomalous Event Attribution**

您是一个知识渊博且乐于助人的金融领域专家，需要基于<时间>和<内容>信息回答用户问题，给出令用户满意的答复。

你需要在下面的内容数据中识别出哪些内容导致了汉邦高科股票在2025-03-28跌停。

| 内容ID | 内容时间 | 内容详情 |
|---|---|---|
| 1 | 2024年8月13日 | 汉邦高科公告拟以0元转让全资子公司天津普泰国信科技100%股权给黄素荣、杨德福。此举旨在降低亏损、优化资产结构，完成后天津普泰将不再纳入公司合并报表范围... |
| 2 | 2025年2月26日 | 声迅股份股价跌破布林线中轨，引发市场关注。历史数据显示该信号出现56次，下跌概率不高。"仓老师"提醒投资者控制仓位，规避潜在风险... |
| 3 | 2025年3月27日 | 在岸人民币兑美元收盘报7.2388，较上一交易日下降100点... |
| 4 | 2025年3月27日 | 汉邦高科公告拟发行股份购买安徽驿路微行科技51%股权，并向实控人李柠控制的北京智耘贰零科技发行股份募集配套资金。公司股票于3月28日起复牌... |
| 5 | 2025年3月10日 | 计算机是现代高速计算电子设备，具备数值与逻辑运算、存储记忆功能。由硬件和软件组成，可分为超级、工业、网络、个人、嵌入式等类型...未来趋势为超高速、小型化、智能化... |
| 6 | 2025年3月18日 | 美股机构配置出现历史性下调，基金经理对美股净低配达23%，降幅创纪录。特朗普政策、贸易战担忧加剧市场悲观情绪，资金加速流向欧洲市场... |
| 7 | 2024年3月14日 | 文章分析汉邦高科在安防监控领域的优势，涵盖业务模式、财务数据、行业前景等，认为其具有较高投资价值，值得长期持有...相关企业信息显示公司注册地址位在北京海淀区.. |
| 8 | 2025年3月9日 | 大盘升值趋势受宏观经济、行业发展、政策环境、资金面及国际形势影响。科技、新能源具高增长潜力；政策与外资动向对市场有显著引导作用... |
| 9 | 2024年2月28日 | 汉邦高科当日下跌5.06%，报4.50元/股，成交3726万元。主力净流出271万元。公司主营安防视频监控产品。三季报显示营收6,878.82万元，净利润亏损4,861.89万元... |
| 10 | 2024年2月23日 | 汉邦高科上涨5.41%，报4.29元/股，成交3326万元。主力净流出118万元。公司主营业务为数字视频监控产品研发销售。财报显示前三季度营收微增，净亏损收窄... |
| 11 | - | 今日公司复牌，存在利好离地、资金出逃的可能... |
| 12 | 2025年2月7日 | 达华智能收盘上涨0.82%，近5日累计涨12.41%。主力净流出564万元，但超大单流入550万元。前三季度营收14.56亿元，净利同比增长238.5%... |
| 13 | 2025年3月18日 | 同为股份在互动平台表示，"云+视频监控"是公司重点发展方向，已拥有完整云端解决方案、覆盖账户系统、视频存储、AI计算等，服务全球客户... |
| 14 | 2025年3月22日 | 财务资助常见形式包括贷款、股权融资、捐赠、政府补贴和债券发行。各类方式在资金来源、偿还要求、资源整合等方面各有特点，企业应根据自身情况选择合适方式... |
| 15 | 2025年3月11日 | 网络不给力，请稍后重试。返回首页... |
| 16 | 2025年3月1日 | 思维列控关联公司南京泰通中标广湛铁路GSM-R无线基站设备项目，金额1920.3万元，招标人为中国铁路设计集团。思维列控持股比例为1.71%... |
| 17 | 2024年3月17日 | 汉邦高科被视作新材料领域龙头，产品应用于电子、光伏、汽车等行业。受益于清洁能源发展，市场需求快速增长，投资者对其未来发展抱有高度期待... |

输出格式要求：请以纯文本形式输出 JSON，不要包含任何代码块标记（如 ```json 或 ```）：
{ "相关新闻序号": [<请填写你的分析结果>], "具体分析": "<请填写你的分析过程>" }

🤖 {
  "相关新闻序号": [4,11],
  "具体分析": "汉邦高科于2025年3月28日复牌后跌停，主因是3月27日公告拟收购安徽驿路微行51%股权并配套融资（内容4），但市场可能视此为"利好出尽"或担忧资产质量与利益输送。叠加内容11提示的"复牌后资金出逃"风险，引发抛售导致跌停。"
  丶

Figure 15: An example instance from the Anomalous Event Attribution.

**Emotion Recognition**

你是一个专业的金融情绪分析助手，你的任务是根据用户提出的问题及其相关的背景信息，来判断用户的当前情绪状态。请从以下几种情绪类型中选择最符合的一项或多项：

**[情绪类型]**

**一、乐观**
- 特点：对未来充满信心，相信市场或投资能够带来积极结果。
- 示例场景：
1. 投资者看到某家科技公司发布了一项突破性技术，认为这将推动股价大幅上涨。
2. 市场经济数据连续几个季度显示强劲增长，投资者预期未来几年股市将持续上扬。
3. 新兴市场国家推出了一系列经济改革政策，投资者对该地区的长期发展前景感到乐观。
4. 某行业正处于快速增长阶段，投资者相信自己的投资组合中相关资产会显著增值。
5. 投资者参加了一场金融论坛，听到多位专家对未来的正面预测后更加坚信市场潜力。

**二、焦虑**
- 特点：对不确定性感到不安，担心潜在风险或损失。
- 示例场景：
1. 投资者持有的股票价格短期内波动剧烈，导致其频繁查看账户以确认资产状况。
2. 国际政治局势紧张，投资者担心战争或制裁可能引发全球经济衰退。
3. 央行宣布即将加息，投资者担忧利率上升会对债券和房地产投资造成负面影响。
4. 某个关键行业的龙头企业突然宣布业绩下滑，投资者对其相关资产的前景感到忧虑。
5. 市场传言某国可能实施资本管制，投资者担心资金无法顺利撤回。

**三、消极**
- 特点：对市场或投资结果产生失望、沮丧或悲观的情绪。
- 示例场景：
1. 投资者长期持有的一只股票因公司丑闻而暴跌，导致其亏损严重。
2. 全球经济进入衰退周期，投资者看到自己的投资组合价值持续缩水。
3. 某行业受到政策监管加强的影响，投资者对该行业的未来失去信心。
4. 投资者尝试了多种策略但均未取得理想收益，开始怀疑自己的投资能力。
5. 市场长期处于低迷状态，投资者对任何新的投资机会都提不起兴趣。

**四、兴奋**
- 特点：因市场上涨或投资成功而感到愉悦和满足。
- 示例场景：
1. 投资者买入的一只股票因公司财报超预期而涨停，当天收益达到两位数。
2. 某热门板块突然成为市场焦点，投资者持有的相关资产迅速升值。
3. 投资者通过精准判断抓住了一次短期交易机会，获得超额回报。
4. 市场因利好消息出现普涨行情，投资者发现自己几乎所有资产都在增值。
5. 投资者参与的新股申购中签，并在上市首日获得了高额收益。

**五、冷静**
- 特点：保持中立和理性，不受市场波动或情绪干扰。
- 示例场景：
1. 市场因突发新闻出现短暂下跌，但投资者并未恐慌抛售，而是选择继续观察。
2. 投资者在市场剧烈波动时坚持既定的投资计划，不轻易调整策略。
3. 面对市场热点板块的快速上涨，投资者并未盲目追涨，而是等待更好的入场时机。
4. 投资者在经济数据不佳时仍保持信心，认为这只是短期现象而非趋势性变化。
5. 即使市场出现重大事件，投资者也能基于数据分析做出客观判断，避免情绪化决策。

**六、后悔**
- 特点：对过去的决策感到遗憾或自责，尤其是当结果不如预期时。
- 示例场景：
1. 投资者因犹豫不决错过了某只股票的最佳买入时机，事后发现该股票大幅上涨。
2. 投资者在市场高位时卖出资产，随后市场继续上涨，导致错失更多收益。
3. 投资者未能及时止损，导致原本的小幅亏损变成重大损失。
4. 投资者听信他人建议进行了一次高风险投资，最终失败并造成巨大损失。
5. 投资者在市场下跌时被迫平仓，事后发现市场很快反弹，导致错失恢复机会。

**[背景信息和用户问句]**
{"市场环境": {
    "大盘走势分析": {"走势表现": {
        "三大指数表现": "上证指数微涨0.3%，深证成指上涨0.5%，创业板指上涨0.7%，市场呈现温和回升态势",
        "技术指标": "上证指数MA5上穿MA10形成金叉，MACD红柱初现，KDJ三线向上发散"},
      "题材分化": { "指数分化": "中证1000指数领涨0.8%，创业板50指数上涨1.2%，显示成长股表现活跃",
        "板块热点": "纺织服装板块涨幅居前，多只个股创年内新高"},
    "市场情绪": {
      "量能变化": "成交量温和放大至1.2万亿，量价配合良好",
      "个股涨跌": "上涨家数增至2500家，涨停家数扩大至35家",
      "主力资金": "北向资金连续三日净流入，单日净买入达50亿元"},
    "综合分析": "市场呈现结构性机会，纺织制造等细分领域显现资金聚集效应"}},
  "用户持股信息": {
  "用户自选股": {
    "公司名称": ["嘉麟杰", "中国平安", "招商银行"],
    "股票代码": ["002486", "601318", "600036"]
  },
  "相关的技术面分析": {
    "嘉麟杰": {
    "趋势分析": {"主要趋势": "周线级别突破下降通道，月线MACD即将金叉",
      "趋势强度和持续性": "布林带开口扩大，ADX指标升至35显示趋势强化"},
    "撑压分析": {"支撑位": "2.80元平台支撑坚实",
      "价格区间": "量能配合下有望挑战3.50元前高"},
    "K线形态分析": {"技术形态": "杯柄形态构筑完成",
      "K线形态": "连续三日放量阳线突破整理平台"
    },
    "量价分析": {
      "成交量和价格": "量能阶梯式放大，OBV能量潮创年内新高",
      "市场情绪": "融资余额单周增长15%，市场关注度显著提升"
    },
    "均线分析": {
      "均线趋向": "5周均线上穿60周均线形成黄金交叉",
      "均线离散": "均线系统呈多头排列"
    },
    "技术指标": {
      "MACD": "日线级别二次翻红，周线级别绿柱缩短",
      "KDJ": "J值维持在80以上超买区域达5个交易日"
    }
    }
  },
  "相关的最新新闻": [
    {
      "日期": "2025-01-02",
      "标题": "嘉麟杰上调回购价格上限至5.2元彰显信心",
      "摘要": "公司宣布将股份回购价格上限大幅提升32%，彰显管理层对企业未来发展强烈信心"
    },
    {
      "日期": "2024-12-30",
      "标题": "纺织行业景气度持续回升",
      "摘要": "海关总署数据显示12月纺织品出口同比大增18%，行业迎来量价齐升周期"
    }
  ],
  "相关的最新研报": [],
  "用户问句": "嘉麟杰这个量价配合，是不是有可能走出一波主升浪？"
}

请严格按照以下JSON格式输出你的分析结果：
  { "cot": "<请在此处填写你的推理过程或判断依据，描述清楚得出结论的理由>", "result": ["<请填写你的情绪分类结果1>","<请填写你的情绪分类结果2>"] }

{
    "cot": "用户当前关注嘉麟杰的量价配合情况，从技术面分析、量能放大和突破形态上推测其可能进入主升浪阶段。结合嘉麟杰近期的回购消息、行业景气度提升以及用户已经持有该股票并浮盈8%的情况，用户对该股票的未来走势可能持乐观态度。此外，从用户积极提问并寻找市场机会的行为来看，其情绪可能受到一定程度的兴奋驱动。",
    "result": ["乐观", "兴奋"]
}

Figure 16: An example instance from the Emotion Recognition.

**Stock Price Prediction**

你是一个金融领域专业分析师。今天是2025-02-14，你要根据用户给出的前一个季度左右的<公司股价数据><涨跌幅数据>和<近期公司相关新闻资讯>，预测今日的股票涨跌情况。

你需要输出一个json，格式为{"涨跌情况":"涨或者跌", "涨跌原因":"分析原因"}。

以下是用户给出的信息：
<公司股价数据>
取数问句: 贵州茅台前90天每天的收盘价
取数结果: 为您找到1条数据

| 股票代码 | 股票简称 | 收盘价 20240926 | 收盘价 20240927 | 收盘价 20240930 | 收盘价 20241008 | ... | 收盘价 20250207 | 收盘价 20250210 | 收盘价 20250211 | 收盘价 20250212 | 收盘价 20250213 |
|---|---|---|---|---|---|---|---|---|---|---|---|
| 600519.SH | 贵州茅台 | 1529.00元 | 1629.20元 | 1748.00元 | 1723.00元 | ... | 1436.00元 | 1431.51元 | 1418.00元 | 1443.00元 | 1465.06元 |

</公司股价数据>

<涨跌幅数据>
取数问句: 贵州茅台前90天每天的涨跌幅
取数结果: 为您找到1条数据

| 股票代码 | 股票简称 | 涨跌幅 20240926 | 涨跌幅 20240927 | 涨跌幅 20240930 | 涨跌幅 20241008 | ... | 涨跌幅 20250207 | 涨跌幅 20250210 | 涨跌幅 20250211 | 涨跌幅 20250212 | 涨跌幅 20250213 |
|---|---|---|---|---|---|---|---|---|---|---|---|
| 600519.SH | 贵州茅台 | 9.29% | 6.55% | 7.29% | -1.43% | ... | 1.64% | -0.31% | -0.94% | 1.76% | 1.53% |

</涨跌幅数据>

<近期公司相关新闻资讯>
标题: 贵州茅台，出手回购10亿元！贵州茅台_新浪财经_新浪网
时间: 2025-02-08
摘要: （文/朱道义 编辑/张广凯）近期股价持续震荡下跌之际，贵州茅台（600519）于日前发布了回购股份实施最新进展公告。
公告显示，2025年1月，贵州茅台通过集中竞价交易方式已累计回购股份68.51万股，占公司总股本的比例为0.0545%，购买的最高价为1507.41元/股、最低价为1436.48元/股，已支付的总金额999909571.13元（不含交易费用）。此次回购符合相关法律法规及公司披露的回购股份方案。
贵州茅台的回购方案，首次披露于2024年9月20日。贵州茅台曾表示，为维护公司及广大投……

标题: 茅台连发两条重要公告！什么情况？- 新浪财经
时间: 2025-01-02
摘要: 1月2日，贵州茅台（600519.SH，股价1488元，市值18692.22亿元）公告，经公司初步核算，2024年度，公司生产茅台酒基酒约5.63万吨，系列酒基酒约4.81万吨；预计实现营业总收入约1738亿元（其中茅台酒营业收入约1458亿元，系列酒营业收入约246亿元），同比增长约15.44%；预计实现归属于上市公司股东的净利润约857亿元，同比增长约14.67%。
另外，贵州茅台还公告，2025年1月2日，公司通过集中竞价交易方式首次回购股份200，900股，已回购股份占公司总股本的比例……
……

</近期公司相关新闻资讯>

{
    "涨跌情况": "涨",
    "涨跌原因": "贵州茅台近期发布了2024年度的生产经营情况，显示公司实现了营业总收入约1738亿元，同比增长约15.44%，归母净利润约857亿元，同比增长约14.67%，业绩表现良好，符合市场预期。此外，公司在2025年1月进行了股份回购，累计回购股份68.51万股，支付总金额近10亿元，显示公司对自身发展的信心，有助于提升市场信心。综合这些积极因素，预计今日股价将上涨。"
}

Figure 17: An example instance from the Stock Price Prediction.

**Question:**

陈先生将100000元存入银行，年利率为1.5%，2年后，他将获得多少元利息? A: 3000 B: 23173 C: 27754 D: 10943

**Answer:** A

(A) An example from the numerical calculation dataset in Fin-Eva Version 1.0

**Question:**

你是一位经验丰富的金融数据分析师，请根据以下<数据参考>，回答出<用户问题>，并给出你的理由。你需要以指定的<输出格式>回答。

### 数据参考

编号: 1

取数问句: 百度2024年8月每日的收盘价

| 股票代码 | 股票简称 | 收盘价
20240801 | …… | 收盘价
20240830 |
|---|---|---|---|---|
| BIDU.O | 百度 | 86.42美元 | …… | 84.62美元 |

编号: 2

取数问句: 百度2024年9月每日的收盘价 ……

### 用户问题

question: 在2024年8月至2024年9月期间，百度的收盘价大于90.00美元的天数有几天?

### 输出格式

……

**Answer:** 6

(B) An example from the numerical calculation dataset in BizFinBench

Figure 18: An Chinese Version of the Comparison of Numerical Calculation Questions in Fin-Eva Team (2023) and BizFinBench