# OpenReview forum: "BizFinBench: A Business-Driven Real-World Financial Benchmark for Evaluating LLMs"
_ICLR.cc/2026/Conference — Submitted to ICLR 2026_

### Official Review · Reviewer_WDtx · 2025-10-26

**Soundness:** 2
**Presentation:** 3
**Contribution:** 3
**Rating:** 4
**Confidence:** 5

**Summary:**

This paper introduces BizFinBench, a business-driven financial benchmark designed to evaluate large language models (LLMs) in real-world Chinese financial scenarios. It comprises 7,605 annotated queries across five dimensions—numerical calculation, reasoning, information extraction, prediction recognition, and knowledge-based QA—grouped into nine fine-grained categories. The authors also propose IteraJudge, an iterative LLM evaluation method that reduces bias in objective metrics. Evaluating 30 proprietary and open-source models, the study reveals that no single model dominates across all tasks, with proprietary models generally outperforming open-source ones, especially in reasoning. The benchmark emphasizes practical applicability, adversarial robustness, and complex reasoning, providing a rigorous tool for future financial AI development.

**Strengths:**

1. Presents a high-quality financial evaluation dataset that demands deep analytical capabilities. The construction process effectively combines expert knowledge with LLM-powered automation, and incorporates adversarial noise injection, ensuring both the high quality and challenging nature of the benchmark.
2. Conducts a comprehensive benchmark evaluation, covering a wide range of leading proprietary and open-source LLMs.
3. The dataset is distinguished by its strong real-world relevance, being sourced from authentic user queries and spanning diverse financial subdomains.

**Weaknesses:**

1. Lack of validation for methodological effectiveness: The paper lacks experiments demonstrating the impact of various data construction steps on data quality. For instance, while the IteraJudge method is claimed to enhance the LLM-as-a-Judge approach, there is no ablation study quantifying how its inclusion or exclusion affects the quality of the experimental results. Similarly, the introduction of adversarial data is not backed by quantitative evidence showing the extent to which it increases task difficulty or how its absence would impact the assessment of model capabilities. This omission makes it difficult to ascertain the necessity of these proposed steps.
2. Potential bias in data sources: Although the paper states that the data originates from real user queries on "Platform A," it does not specify whether these queries cover different types of financial institutions (e.g., banks, brokerages, insurance companies) or diverse user groups (including institutional investors). The distribution across these user types is also unclear. This lack of detail raises concerns about potential discrepancies between the benchmark's problem distribution and that of real-world financial scenarios, possibly limiting its representativeness.
3. Insufficient model comparison: While the evaluation includes many mainstream large language models, it overlooks several specialized models developed specifically for the financial domain, such as Fin-R1[1] and Dianjin-R1[2]. Given that BizFinBench is proposed as a financial evaluation benchmark, the absence of comparisons with these domain-specific models is a significant shortcoming.

[1]Liu Z, Guo X, Lou F, et al. Fin-r1: A large language model for financial reasoning through reinforcement learning[J]. arXiv preprint arXiv:2503.16252, 2025.

[2]Zhu J, Chen Q, Dou H, et al. Dianjin-r1: Evaluating and enhancing financial reasoning in large language models[J]. arXiv preprint arXiv:2504.15716, 2025.

**Questions:**

1. What is the rationale for selecting GPT-4o for ITERAJUDGE, rather than a potentially more capable alternative? Could the use of a single model for evaluation introduce or amplify specific model biases within the assessment results?
2. How is the correctness and effectiveness of the data filtering steps described in the paper demonstrated, and what specific impact do these steps have on the final data quality?
3. How does the single-platform sourcing of query data ensure comprehensive coverage of question-answering needs across the entire financial domain, and what measures guarantee that the problem distribution accurately reflects real-world scenarios?
4. The paper claims that BizFinBench is "the first benchmark specifically designed to evaluate LLMs in real-world financial applications." However, the previously published CCC dataset from Dianjin-R1[2] is also constructed from real-world Chinese financial customer-service dialogues.

(1)	What are the key distinctions between BizFinBench and the CCC dataset?

(2)	What are the specific advantages of the data in this work?

(3)	If the advantages are primarily in data quality or question diversity, and both datasets are derived from real-world financial data, is the claim of being "the first" justified?

Suggestion
1. It is recommended to supplement the performance evaluation by including large language models that are specifically designed for the financial domain.

---

### Official Review · Reviewer_v4JY · 2025-10-27

**Soundness:** 2
**Presentation:** 1
**Contribution:** 2
**Rating:** 2
**Confidence:** 4

**Summary:**

This paper introduces BizFinBench, a benchmark designed for real-world business-oriented question answering in the financial domain. It consists of a dataset deeply integrated with realistic financial scenarios and a multi-dimensional evaluation framework called IteraJudge, which is based on iterative calibration via LLM-as-a-Judge. The benchmark covers five dimensions and nine fine-grained tasks. The authors also report evaluation results on 30 models and provide analysis based on the findings.

**Strengths:**

1. The dataset is sourced directly from trading platforms with large user bases, which ensures its grounding in real-world financial business scenarios.

2. The paper proposes IteraJudge, an LLM-as-a-Judge evaluation method capable of providing reliable and multi-dimensional assessments for complex problems.

**Weaknesses:**

1. While the paper claims that BizFinBench features contextual complexity and adversarial robustness, it does not provide any statistical metrics to quantify such complexity, nor does it include examples illustrating how noise or adversarial elements are introduced into the data.

2. The presentation of related work contains notable omissions in both form and content:

    (1) In Table 1, the "task" column omits Financial numerical reasoning, despite this being explicitly included in the paper's own taxonomy of nine task types.

    (2) The "except" label in the "source" column of Table 1 lacks a clear and adequate explanation.

    (3) Several recently released and relevant datasets are missing from the comparison, such as:
DocMath-Eval (https://aclanthology.org/2024.acl-long.852v2.pdf);
FinanceMath (https://aclanthology.org/2024.acl-long.693v2.pdf);
FinanceReasoning (https://aclanthology.org/2025.acl-long.766/).
These works should be included for a more complete comparison.

3. The appendix suffers from inconsistent formatting. Furthermore, although BizFinBench is a fully Chinese dataset, English versions of the questions and prompts should also be provided, especially since the footnote on page 5 implies their availability. This inconsistency is confusing.

4. Despite the detailed experimental results, the accompanying analysis and insights are relatively limited:

    (1) The trend that model performance improves with increased parameter size does not consistently hold. For example, the Qwen2.5-VL series shows inconsistent behavior on ER and SP tasks.

    (2) The third insight regarding distilled models lacks novelty; more distinctive analytical perspectives would be valuable.

    (3) The local reasoning models Qwen3-235B-A22B-Thinking and Deepseek-R1 perform significantly worse than their instruction-tuned or general-purpose counterparts (Qwen3-235B-A22B-Instruct and Deepseek-V3), which contradicts both the proprietary model trend and general expectations. This discrepancy may indicate potential quality issues within the dataset.

5. There are multiple spelling and grammatical issues:

    (1) In Figure 2, "Calcilation" should be corrected to "Calculation".

    (2) The annotation in Figure 1 uses "An Chinese" instead of the correct "A Chinese".

    (3) In Section 4.3, the sentence "All results as shown in Table 3" lacks a verb and is grammatically incomplete.

6. Although examples are provided for all nine task types in the appendix, the examples for FNC and FTR are overly similar. Both merely involve data lookup from tables without any additional reasoning components. As such, they fail to demonstrate the distinction between the two task types.

**Questions:**

1. The contents of Figure 3 appear inconsistent with the actual construction of BizFinBench. Given that the dataset is derived from processed data collected from trading platforms, it is unclear why "Academic" would be listed as one of the sources. Could the authors clarify this?

Additionally, the authors are encouraged to address the issues raised in the Weaknesses section.

---

### Official Review · Reviewer_tPWb · 2025-10-30

**Soundness:** 3
**Presentation:** 3
**Contribution:** 2
**Rating:** 4
**Confidence:** 4

**Summary:**

The paper proposes BizFinBench, a finance-focused benchmark in Chinese containing 7,605 instances. The queries are harvested from real-users via an online platform and filtered/classified with GPT-4o.

The paper evaluates a wide array of open and closed LLMs, including GPT, Gemini, Qwen2.5, DeepSeek, and Llama. The findings are straightforward: proprietary models dominate, with a few exceptions by DeepSeek-R1 or Qwen-235B.

The paper's biggest contribution is the dataset itself.

**Strengths:**

(1) The paper presents BizFinBench a Chinese financial benchmark consisting of original prompts from actual users. This differs from the vast-majority of synthetic datasets flooding recently.

(2) The paper uses LLM-Judges for evaluation, which is backed by comparing different LLM-Judges. However, this could have been better by comparing with a human baseline.

**Weaknesses:**

(1) Some subsets of the dataset are already highly saturated with top-performing models scoring over 90.

(2) The contribution (dataset) is limited to a small sub-group of NLP, the Chinese community interested in Finance.

(3) The paper would be better with an error analysis that provides a comparison with closed and open models. This will provide better guidance on what model makers need to improve.

(4) The evaluation setup says maximum token is limited to 1024 tokens, Im not sure if this enough especially for reasoning models like R1. Also evaluation would be better if it was repeated and the average was reported.

In general while the dataset is a valuable contribution considering the recent flood of benchmarks, and its narrow topic, i think it might be less interesting to the wider community.

**Questions:**

See weakness.

---

### Official Review · Reviewer_wE8m · 2025-11-01

**Soundness:** 2
**Presentation:** 2
**Contribution:** 2
**Rating:** 2
**Confidence:** 5

**Summary:**

The paper presents BizFinBench, a business-driven Chinese financial benchmark designed to evaluate LLMs in real-world financial scenarios. The dataset comprises 7,605 annotated QAs that span five key dimensions, divided into nine task categories. The authors also introduce IteraJudge, an iterative, multi-dimensional evaluation approach that aims to reduce LLM-based judge bias. Thirty LLMs (proprietary and open-source) are evaluated on BizFinBench, and extensive results highlight both progress and gaps in current models’ abilities on challenging financial tasks.

**Strengths:**

1. The authors provided a comprehensive 7,605 annotated instances across 5 major financial dimensions and 9 granular categories, ensuring broad domain and task coverage.
2. The authors conducted extensive experiments, evaluating 30 close and open source models in total.
3. The authors proposed a new evaluation method, IteraJudge, to reduce bias during evaluation.

**Weaknesses:**

1. Existing works[1, 2] should be discussed in the related work section.
2. Task difficulty: Overall, the goal of a benchmark is to facilitate the development and improvement of future models. However, in its current form, this benchmark is not sufficiently challenging. Current models perform very well on 5 out of 9 tasks (the SOTA model achieving over 90 on 4 tasks and 87 on another), raising main concerns about the dataset’s usefulness.
3. It is not clear what current models failed; any error analysis may help to illustrate this.
4. In Lines 196–197, the authors claim that “BizFinBench is the only benchmark explicitly designed around real-world financial operations and user interactions.” However, based on the reviewer’s knowledge, [2] also introduces real-world scenarios. This raises the question of how BizFinBench differs from [2] in terms of design goals and real-world applicability.



[1] Jiang, Junzhe et al. “FinMaster: A Holistic Benchmark for Mastering Full-Pipeline Financial Workflows with LLMs.” ArXiv abs/2505.13533 (2025): n. pag.
[2] Xie, Qianqian et al. “FinBen: A Holistic Financial Benchmark for Large Language Models.” Neural Information Processing Systems (2024).

**Questions:**

Please check the Weaknesses section.

---

### Meta-Review · Area_Chair_5t9f · 2026-01-08

**Summary:**

This work presents a potentially valuable resource focused on real-world financial tasks. The construction process effectively combines expert knowledge with LLM-powered automation and incorporates adversarial noise injection. However, the reviewers raised several consistent concerns that limit the paper’s overall impact.

First, multiple reviewers questioned the difficulty and usefulness of the benchmark. A large portion of the tasks appears to be saturated, with current state-of-the-art models achieving very high performance on most subsets, which raises doubts about the benchmark’s ability to meaningfully differentiate model capabilities or drive future progress.

Second, reviewers highlighted insufficient analytical depth and statistical rigor. The evaluation primarily reports raw scores without repeated runs, variance analysis, or statistical significance testing, and key experimental choices (e.g., token limits) are not sufficiently justified.

Third, reviewers expressed concerns about incomplete and outdated related work and comparisons. Several important recent benchmarks and datasets in financial reasoning are missing, and the paper does not adequately clarify how BizFinBench differs from existing real-world financial benchmarks in terms of design goals and real-world applicability.

I tend to reject this paper.

**Reviewer Concerns:**

The author do not provide rebuttal.

**Reviewer Scores:**

The author do not provide rebuttal, so I think the reviewer will keep their origin score.

Reviewer wE8m: score 2

Reviewer tPWb: score 4

Reviewer v4JY: score 2

Reviewer WDtx: score 4

---

### Decision · Program_Chairs · 2026-01-26

Reject